# Hepatic Steatosis and Fibrosis in Chronic Inflammatory Bowel Disease

**DOI:** 10.3390/jcm11092623

**Published:** 2022-05-06

**Authors:** Claudia Veltkamp, Shuai Lan, Eleni Korompoki, Karl-Heinz Weiss, Hartmut Schmidt, Helmut K. Seitz

**Affiliations:** 1Department of Gastroenterology, Hepatology and Transplantation Medicine, University Hospital Essen, Hufelandstraße 55, 45147 Essen, Germany; hartmut.schmidt@uk-essen.de; 2Department of Internal Medicine, Salem Hospital, 69121 Heidelberg, Germany; shuailan@hotmail.com (S.L.); karlheinz.weiss@stadtmission-hd.de (K.-H.W.); helmut_karl.seitz@urz.uni-heidelberg.de (H.K.S.); 3Department of Clinical Therapeutics, Alexandra Hospital, National and Kapodistrian University of Athens, 157 72 Athens, Greece; e.korompoki@imperial.ac.uk; 4Centre of Liver- and Alcohol Diseases, Ethianum Clinic, 69115 Heidelberg, Germany; 5Faculty of Medicine, University of Heidelberg, 69117 Heidelberg, Germany

**Keywords:** hepatic steatosis, hepatic fibrosis, Crohn´s disease, ulcerative colitis, elastography

## Abstract

Background and Purpose: Chronic inflammatory bowel diseases (IBD) frequently affect extraintestinal organs including the liver. Since limited evidence suggests the presence of liver disease in IBD patients, we studied the frequency of hepatic steatosis and fibrosis in these patients and characterized disease-related factors. Methods: In this retrospective, cross-sectional, hospital-based, single-center study, consecutive patients with Crohn’s disease (CD) and ulcerative colitis (UC) were included who had undergone routine abdominal ultrasound including transhepatic elastography. Hepatic steatosis was diagnosed by hyperechogenicity on B-mode ultrasound and by measuring controlled attenuation parameter (CAP). Hepatic fibrosis was assumed if transhepatic elastography yielded a stiffness > 7 kPa. Results: 132 patients (60% CD) with a median disease duration of 10 years were included. Steatosis assessed by B-mode ultrasound and CAP correlated well. Of the IBD patients, 30.3% had non-alcoholic fatty liver (NAFL). Factors associated with NAFL were age, BMI, duration of disease, as well as serum activities of aspartate-aminotransferase (AST) and gamma-glutamyl-transpeptidase (GGT). In multivariate analysis, only disease duration was independently associated with hepatic steatosis. Hepatic fibrosis was found in 10 (8%) of all IBD patients, predominantly in patients with CD (10/11). Conclusions: Pure hepatic steatosis is common in both CD and UC, whereas hepatic fibrosis occurs predominantly in CD patients. Association of disease duration with NAFLD suggests a contribution of IBD-related pathogenetic factors. Longitudinal studies are needed to better understand the impact of IBD on hepatic disorders.

## 1. Introduction

Chronic inflammatory bowel diseases (IBD) comprising ulcerative colitis (UC) and Crohn’s disease (CD) result from a dysregulated immune response to the enteric resident microbiota in genetically susceptible hosts [1]. Beyond the primary intestinal manifestation, about 40% of patients with IBD have extraintestinal manifestations [2,3], and among these, primary sclerosing cholangitis (PSC) is particularly common [4,5]. Moreover, there is increasing awareness of a potential association of IBD with other liver disorders including so-called non-alcoholic fatty liver disease (NAFLD), which comprises pure fatty liver (NAFL) and steatohepatitis (NASH) with various degrees of fibrosis. In 10 to 25% of patients, NAFL progresses to NASH [6]. NASH is associated with hepatic fibrosis and cirrhosis, liver failure and hepatocellular carcinoma [7,8]. The factors facilitating progression from NAFL to NASH are still incompletely understood in the general population, and even less is known in IBD patients. Several studies suggest that patients with IBD may be at an increased risk for NAFLD as a result of chronic inflammation, changes in the microbial flora with increased uptake of endotoxins from the intestine to the liver, and dietary or digestive alterations [9]. Moreover, several drugs that are routinely used for treatment of IBD, including corticosteroids, azathioprine/6-mercaptopurine, methotrexate and anti-tumor necrosis factor alpha (TNF), have potential hepatotoxic side effects. Hence, better insight into the factors leading to the development of hepatic steatosis and fibrosis in IBD patients is needed. 

Previous studies reported a wide range for the prevalence of hepatic steatosis, of 1.5% to 55%, in cohorts of IBD patients [4]. Part of this variability may be explained by the choice of diagnostic test. Although liver biopsy is still considered the diagnostic gold standard, its routine use in patients without overt liver disease is prohibitive. B-mode hepatic ultrasound visualizes diffuse hyperechogenicity as a correlate of hepatic steatosis. Transhepatic elastography semi-automatically measures surrogate parameters of hepatic steatosis, including controlled attenuation parameter (CAP). Elastography can also determine liver stiffness, a surrogate marker of liver fibrosis. However, hepatic ultrasound is not part of the routine diagnostic work-up for IBD patients, and available studies have barely used multimodal ultrasound techniques to describe the spectrum of liver manifestations of IBD.

In the present study, we examined the frequency of hepatic steatosis and hepatic fibrosis in a consecutive, single-center cohort of IBD patients using both standard hepatic ultrasound and hepatic elastography. We also explored which clinical and laboratory parameters are associated with hepatic alterations. 

## 2. Material and Methods

We performed a retrospective, single-center cross-sectional study at the Department of Internal Medicine and Gastroenterology of the Salem Hospital, an academic teaching hospital of the University Heidelberg, Germany. IBD patients were seen at the department after referral by general practitioners and internal medicine physicians. Consecutive patients with an established diagnosis of CD or UC who presented between 2012 and 2016 were identified using the hospital medical records system. 

Patients were eligible for the study only if they had received a hepatic ultrasound. Patients were excluded from the analysis if they had a history of primary liver diseases, including autoimmune hepatitis, primary sclerosing cholangitis, primary biliary cholangitis, autoimmune cholangitis, viral hepatitis, or liver tumors. We also excluded patients with severe heart failure or diabetes (defined as established diagnosis, HbA1c > 6.4%, non-fasting glucose level > 180 mg/dL, or on antidiabetic medication), or if they were chronically using alcohol beyond an estimated amount of 10 g/d for women and 20 g/d for men, respectively.

A standardized, pseudonymized case report form was used to extract data including patient characteristics, body mass index, smoking habits, time interval since first diagnosis of IBD and previous medical treatment of IBD from the medical record. Moreover, results of liver function tests and, if available, serum total cholesterol and triglyceride concentrations were also captured from records.

Abdominal ultrasound was performed as a routine procedure in all study participants by a trained physician using standard procedures established in our department. All ultrasounds were run on a Hitachi Hi Vision Preirus ultrasound scanner using a 3.5 MHz transducer. Liver ultrasound reports were evaluated in relation to liver size with hepatomegaly defined as >14 cm in the midclavicular line.

Liver steatosis was categorized based on the following ultrasound features: grade 1 (mild) with a slight increase in liver echogenicity or slightly impaired hepatorenal echo contrast difference, grade 2 (moderate) with both positive hepatorenal echo contrast and bright liver echogenicity as well as an only obscure visualization of the diaphragm, and grade 3 (severe), a marked increase in hepatic echogenicity, poor or no penetration of the posterior segment of the right lobe of the liver, no visualization of the diaphragm and decreased or no visualization of the hepatic vessels and diaphragm [10]. 

Hepatic elastography with associated controlled attenuation parameter (CAP) was routinely performed as part of hepatic ultrasound at our institution. Hepatic steatosis was measured by transient elastography, and liver stiffness was measured with a fibroscan (Echosens, Paris, France) using the M—probe as described [11]. Briefly, the tip of the transducer probe was placed on the skin between the ribs and above the right lobe of the liver. Measurement depth was between 25 and 65 mm below the skin surface. Ten measurements were performed with a minimum success rate of 70%. Following the recommendation by Saroli [12] and by a recent meta-analysis [13], any grade of NAFL was defined as CAP > 248 dB/m. For measurements of stiffness by elastography, results were expressed in kilo Pascals, and the median value of measurements was determined. Liver fibrosis and cirrhosis were defined as transient elastography (TE) measurement > 7.0 kPa and >12.5 kPa, respectively [14].

## 3. Statistical Analysis

All continuous variables were tested for normal distribution. Continuous variables were expressed as median (25–75 percentile), and categorical variables were presented as numeric (%). Group means of continuous variables were compared by Mann–Whitney test, and categorical variables by χ^2^ to evaluate statistical differences among the different groups. A multivariate logistic regression analysis (Enter method) was performed to examine which variables were associated independently with the sonographic diagnosis of steatosis. These included age, BMI, duration of disease, use of steroids, AST, gamma-glutamyl transferase, CRP, and use of nicotine. Similarly, we performed a multivariate logistic regression analysis (Enter method) to examine which variables were associated independently with hepatic fibrosis. These included the following variables: disease duration, AST, gamma-glutamyl transferase, IBD type, hepatomegaly and ALT. A *p* value of less than 0.05 was considered statistically significant. 

The study was approved by the ethics committee of the Medical Faculty of the University Hospital Heidelberg (No S-851-2021). Written informed consent was waived as only a retrospective analysis of routine data was performed. 

## 4. Results

A total of 132 patients were included in the study. Patient characteristics are shown in Table 1. Sixty percent of patients suffered from CD and 40% of patients from UC. Median age of the entire cohort was 42 years. A small majority of patients were female (57%). At presentation to our service, median duration of disease was 10 years. 

Subsequent analyses were based on the cohort with available B-mode ultrasound. B-mode ultrasound found evidence of any hepatic steatosis in 40 of 132 patients (30.3%). Among them, 26 (65%) had a grade 1, 13 (32.5%) had a grade 2, and 1 patient had a grade 3 hepatic steatosis. CAP measurements were available in 89 of the 132 patients. Of these, 31.5% had a CAP value ≥ 248 db/m. Determination of hepatic steatosis by CAP corresponded well to steatosis diagnosed by B mode ultrasound (R: 0.735, *p* = 0.048). Thirty-three percent of the IBD patients had hepatomegaly. Half of the patients with hepatic steatosis also had hepatomegaly (Table 2). 

Patients with hepatic steatosis were significantly older, had a higher body weight and BMI, and had a longer disease duration. Immunosuppressive medication did not significantly affect the presence of steatosis (Table 1). 

In multivariate logistic regression analysis, including age, BMI, duration of disease, AST, GGT, use of steroids, use of nicotine, and CRP, disease duration was the only variable that was independently associated with hepatic steatosis (Appendix A). 

Next, we examined the presence of liver stiffness in IBD patients (Table 3). Our study had a suitable population for the measurement of hepatic stiffness by TE, because the patients were all non-obese, allowing the M probe to be used successfully in all patients. Hepatic elastography showed that the median stiffness in IBD patients was normal. However, 8% (11 of 132) of patients had evidence of liver fibrosis (i.e., stiffness ≥ 7) (Table 2). Most patients (6%) had an F2 grade, 2% had an F3 grade and no patient had an F4 grade (i.e., liver cirrhosis). Patients with fibrosis significantly more frequently had hepatomegaly. 

In multivariate logistic regression analysis, including duration of disease, AST, ALT and GGT, hepatomegaly and type of disease, disease duration was the only variable that was independently associated with hepatic fibrosis (Appendix A). 

Of the 11 IBD patients with increased stiffness, only 7 also had evidence of hepatic steatosis, suggesting that the development of hepatic fibrosis is partially independent of hepatic steatosis. 

Finally, we compared the presence of hepatic manifestations in CD versus UC (Table 4). Patients with CD and UC did not differ significantly in terms of hepatic steatosis, In contrast, a marked difference was evident for hepatic stiffness. Of the 11 patients diagnosed with hepatic fibrosis, 10 suffered from CD. 

## 5. Discussion

We investigated the association of IBD with NAFLD in terms of liver steatosis and liver fibrosis using B-mode ultrasound and TE in a cohort of IBD patients. The key findings of our study are that (1) hepatic steatosis is common in a middle-aged cohort of IBD patients, (2) the development of hepatic steatosis in IBD patients is associated with disease duration, (3) only a small proportion of IBD patients have evidence of hepatic fibrosis, and (4) hepatic fibrosis occurs predominantly in CD patients. 

Previous studies reported a substantial variety for the prevalence of NAFLD in IBD patients and yielded controversial findings when comparing the prevalence of NAFLD in IBD patients against the general population. The population prevalence of NAFLD is 23.7% in Europe and 25.4% globally [6,7,15]. Some investigators found a markedly lower prevalence of NAFLD in IBD patients (7.2%, 8.2%) [16], [17]. Other investigators and a recent systematic review reported a similar or higher prevalence of NAFLD in IBD patients compared to the general population [12,18,19]. The geographical origin of the population also affects the risk of NAFLD in IBD patients. American patients had the highest (43%), European patients an intermediate (31%), and Asian patients the lowest prevalence of NAFLD among IBD patients (13%) [19]. Alternatively, differences in the prevalence of NAFLD among IBD cohorts may result from different diagnostic tests (e.g., liver function tests, ultrasound, MRI, CT, liver biopsy). However, even in studies using biopsy samples, a large variability in the prevalence of NAFLD between 4 and 55% of IBD patients has been reported [4,20]. In our study, findings of B-mode ultrasound and CAP measurements correlated well, supporting that our findings are valid and in line with previous findings by others [21] using B-mode ultrasound in combination with CAP measurement of the liver. 

Our findings support the notion that disease-related factors contribute to pathogenesis of NAFLD in IBD patients. In the general population, the prevalence of NAFLD increases with age, and patients often suffer from diabetes, obesity, hyperlipidemia, hypertension, and chronic kidney disease [18]. In contrast, our IBD patient cohort was relatively young (median age 42 years), had a median BMI of 23, and we excluded patients with a history of diabetes or increased consumption of alcohol. 

In our study, only disease duration was independently associated with NAFLD in multivariate analysis. The mechanisms underlying this disease-associated effect are largely speculative. A major characteristic of IBD is the loss of mucosal integrity, resulting in a “leaky gut” with the uptake of endotoxins. Potential factors in IBD patients include a disrupted mucosal barrier in the gut that no longer protects against the immigration of bacteria of the normal intestinal flora into the intestinal wall. Bacteria-specific immune cells are found in experimental IBD [22], which, together with other intestinal pro-inflammatory factors including bacterial endotoxin, may translocate from the gut into the portal vein and subsequently trigger intrahepatic processes of inflammation. These endotoxins enter the liver and bind at receptors on Kupffer cells, resulting in the secretion of cytokines and chemokines, which activate hepatic stellate cells for fibrogenesis [23]. The larger the intestinal lesions are, the more endotoxins that enter the portal system and may affect the liver. Intriguingly, stabilization of the mucosa by a layer of protective bacteria improved hepatic steatosis in animal models [24] and human subjects [25]. According to the “multiple hit” hypothesis of the pathogenesis of NAFLD [26], several insults acting together are necessary to lead to NAFLD in genetically predisposed patients. Similarly to IBD, there is evidence for substantial changes in microbiota in diverticular diseases of the colon [27]. Colonic diverticular disease is also associated with NAFLD, suggesting that changes in gut microbiota may play a role in the pathogenesis of NAFLD in these conditions [28].

Certain eating habits of IBD patients and malabsorption due to intestinal inflammation may lead to low body weight, which can cause NAFL [29,30]. Conversely, some IBD patients with NAFLD eat an unbalanced, high caloric diet, resulting in a higher BMI [18].

Although several drugs have hepatotoxic side effects, these drugs were not associated with NAFL in our study. Similarly, previous studies found no effect from azathioprine/6-mercaptopurine [17] or methotrexate on the development of liver fibrosis [31]. We did not investigate the role of Sulfasalazine, which may have hepatotoxic effects [32]. Elevated tumor necrosis factor alpha is not only an important cytokine in the pathogenesis of IBD but also one of the main proinflammatory factors involved in the earliest phases of a variety of liver diseases [33]. Indeed, anti-TNF alpha antagonism reversed intestinal inflammation and improved steatosis in rodent models [34] and humans [35].

Transhepatic elastography is a well-established, highly accurate, non-invasive method to measure hepatic fibrosis in the liver [11]. Insights into the prevalence and pathogenesis of hepatic fibrosis in IBD patients are very limited to date. Eight percent of our IBD patients showed increased liver stiffness on hepatic elastography, reflecting moderate fibrosis in accordance with previous studies [12,36,37]. Recent studies using similar threshold levels for F1 to F4 fibrosis with transient elastography found grades of fibrosis in IBD patients similar to those found in our study [14,38]. 

Remarkably, most patients with hepatic fibrosis in our study suffered from CD as compared to UC. This suggests that specific factors other than just any kind of intestinal inflammation lead to hepatic fibrosis, some of which may be more prominent in CD. Such factors may include vitamin D deficiency and bile acid malabsorption. Our study also reveals that hepatic fibrosis develops at least partially independently of hepatic steatosis in CD patients, as more than one third of patients with fibrosis had no evidence of NAFLD. A previous study in IBD patients with NAFLD diagnosed by imaging or biopsy originally reported a rare progression to hepatic fibrosis. However, the degree of fibrosis was only estimated using a NAFLD fibrosis score and not ultrasound, elastography or histology [37]. In the general population, NAFLD may progress to NASH and liver cirrhosis in connection with unfavorable genetic factors [39,40]. Further investigation of genetic and other factors underlying the development and progression of liver fibrosis in IBD and particularly CD patients is warranted. 

The main limitation of our study is the retrospective, cross-sectional study design, which did not allow complete collection of data for several parameters and precluded any follow-up. Because the study was based on data from a single center, the generalizability is limited, and a selection bias is possible. Moreover, we had no detailed information about the duration of exposure to the respective drugs. On the other hand, we prespecified eligibility criteria.

In conclusion, our study highlights that liver steatosis is common in IBD patients and suggests that the inflammatory disease process contributes to the development of fatty liver disease both in CD and UC. In contrast, hepatic fibrosis appears to occur predominantly in CD. While future longitudinal studies are needed to better understand the prognostic impact of IBD on NAFLD and liver fibrosis development, a multimodal hepatic ultrasound may be useful for the diagnostic assessment of liver disease in IBD patients.

## Figures and Tables

**Table 1 jcm-11-02623-t001:** Patient characteristics.

Variables	All Patients (*n* = 132)	No Steatosis (*n* = 92)	Steatosis (*n* = 40)	*p* Value
Age, years	42 (31–57)	40 (28–59)	45 (40–52)	0.001
Gender, males	57 (43)	40 (43)	17 (45)	0.5
Weight, kg	68 (56–80)	64 (54–79)	73 (64–94)	0.02
Height, cm	170 (164–178)	171 (163–178)	170 (166–180)	0.5
BMI, kg/m^2^	23 (20–26)	22 (20–26)	24 (22–29)	0.02
Nicotine	31 (23)	19 (21)	12 (30)	0.17
Type of diagnosis				0.16
Crohn’s disease	79 (60)	52 (57)	27 (68)	
Ulcerative colitis	53 (40)	40 (43)	13 (32)	
Disease duration, years	10 (3–20)	10 (3–17)	11 (3–17)	0.14
AST, IU/L	18 (14–25)	17 (14–23)	19 (16–32)	0.06
ALT, IU/L	17 (11–27)	15 (10–23)	21 (14–37)	0.11
ALP, IU/L	82 (60–100)	78 (56–98)	91 (72–102)	0.07
γ-GT, IU/L	23 (16–51)	23 (15–44)	30 (19–85)	0.06
Total bilirubin, mg/dl	0.3 (0.2–0.5)	0.3 (0.2–0.5)	0.3 (0.2–0.5)	0.6
Cholesterol, mg/dl	139 (109–185)	135 (102–186)	161 (126–187)	0.3
LDL, mg/dl	76 (55–121)	73 (54–121)	85 (56–129)	0.9
Triglycerides, mg/dl	100 (78–142)	90 (77–133)	118 (82–195)	0.06
Steroids	112/127 (88)	80/87 (92)	32/40 (80)	0.054
Azathioprine/Mercapt	39/127 (31)	27/87 (31)	12/40 (30)	0.5
Methotrexate	7/127 (6)	5/87 (6)	2/40 (5)	0.6
Anti-TNF-Antibody	26/127 (20)	20/87 (22)	6/40 (15)	0.21

Categorical variables presented as numeric (%) and continuous as median (25–75 percentile). A *p* value of less than 0.05 was considered statistically significant.: Information about treatment was available in 127 patients.

**Table 2 jcm-11-02623-t002:** Findings of hepatic ultrasound and elastography.

Variables	All Patients (*n* = 132)	No Steatosis (*n* = 94)	Steatosis (*n* = 38)	*p* Value
Hepatomegaly	44 (33)	25 (27)	19 (50)	0.01
CAP (*n* = 89)	219 (183–267)	203 (175–235)	292 (237–316)	<0.001
Stiffness [kPa]	4.3 (3.4–5.3)	4.3 (3.5–4.9)	4.2 (3.3–6.3)	0.4
Number (%) of patients with fibrosis (stiffness ≥ 7)	11 (8)	4 (4)	7 (18)	0.01

Categorical variables presented as numeric (%) and continuous as median (25–75 percentile). Hepatomegaly was defined as >14 cm in the midclavicular line. CAP: Controlled attenuation parameter assessed by hepatic elastography; CAP scores were measured in 89 patients after the CAP module had become available.

**Table 3 jcm-11-02623-t003:** Characteristics of patients according to steatosis and hepatic fibrosis pattern.

Variables	Combined Steatosis and Fibrosis (*n* = 7)	Pts without Combined Steatosis and Fibrosis (*n* = 125)	*p* Value
Age, years	46 (41–48)	41 (31–58)	0.9
Gender, males	1 (14)	56 (45)	0.1
Weight, kg (*n* = 107)	70 (60–89)	67 (56–80)	0.6
Height, cm (*n* = 101)	167 (163–170)	171 (165–179)	0.3
BMI, kg/m^2^ (*n* = 95)	24 (22–33)	23 (20–26)	0.3
Nicotine	1 (14)	30 (24)	0.48
Disease duration, years	30 (17–37)	9 (3–19)	<0.001
Type of diagnosis			0.025
Crohn’s disease	7 (100)	72 (58)	
Ulcerative colitis	0 (0)	53 (42)	
Hepatomegaly	6 (86)	38 (30.4)	0.025
AST, IU/L	43 (35–71)	17 (14–23)	<0.001
ALT, IU/L	42 (23–91)	16 (11–26)	0.003
ALP, IU/L	82 (61–104)	82 (60–100)	0.8
γ-GT, IU/L	88 (47–124)	23 (15–45)	0.005
Total bilirubin, mg/dL	0.5 (0.4–0.9)	0.3 (0.2–0.4)	0.07
Steroids	6/7 (86)	106/120 (88)	0.6
Azathioprine/Mercaptopurine	2/7 (29)	37/120 (31)	0.6
Methotrexate	0/7 (0)	7/120 (6)	0.7
Anti-TNF	1/7 (14)	25/120 (21)	0.6

Categorical variables presented as numeric (%) and continuous as median (25–75 percentile). Hepatomegaly was defined as >14 am in the midclavicular line, CAP: Controlled attenuation parameter assessed by hepatic elastography.

**Table 4 jcm-11-02623-t004:** Characteristics of patients according to type of diagnosis (Crohn‘s disease versus ulcerative colitis).

Variables	Crohn’s Disease (*n* = 79)	Ulcerative Colitis (*n* = 53)	*p* Value
Age, years	44 (33–53)	40 (30–62)	0.9
Gender, males	31 (39)	26 (49)	0.17
Weight, kg	64 (54–79)	73 (64–94)	0.054
Height, cm	170 (163–176)	173 (167–181)	0.2
BMI, kg/m^2^	23 (20–27)	23 (20–25)	0.7
Nicotine	25 (32)	6 (11)	0.005
Disease duration, years	9.5 (3–21.3)	10 (3–15)	0.27
Hepatomegaly	29 (37)	15 (28)	0.15
Steatosis (sonography)	27 (34)	13 (16)	0.16
CAP	222 (186–277)	205 (174–255)	0.31
Stiffness [kPa]	4.3 (3.5–5.3)	4.1 (3.4–5.2)	0.47
Fibrosis (Stiffness ≥ 7)	10 (13)	1 (2)	0.025
AST, IU/L	19 (15–26)	17 (14–23)	0.2
ALT, IU/L	17 (10–28)	15 (11–24)	0.11
ALP, IU/L	83 (64–99)	80 (53–103)	0.33
γ-GT, IU/L	23 (14–47)	24 (16–60)	0.6
Total bilirubin, mg/dL	0.3 (0.2–0.4)	0.3 (0.2–0.5)	0.3
Cholesterol, mg/dL	130 (108–171)	175 (142–201)	0.02
LDL, mg/dL	63 (54–85)	115 (98–136)	0.009
Triglycerides, mg/dL	90 (75–145)	114 (78–136)	0.94
Steroids	67/76 (88)	45/51 (88)	0.6
Azathioprine/Mercaptopurine	26/76 (34)	13/51 (25)	2
Methotrexate	5/76 (7)	2/51 (4)	0.4
Anti-TNF	19/76 (25)	7/51 (14)	0.09

Categorical variables presented as numeric (%) and continuous as median (25–75 percentile). Hepatomegaly was defined as >14 cm in the midclavicular line. CAP: Controlled attenuation parameter assessed by hepatic elastography; CAP scores were measured in 89 patients after the CAP module had become available.

## Data Availability

The data presented in this study are available on request from the corresponding author. The data are not publicly available due to waived consent.

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
