# Peer review of "Hepatic Steatosis and Fibrosis in Chronic Inflammatory Bowel Disease"

_jcm, 2022, doi:10.3390/jcm11092623_

Round 1

Reviewer 1 Report

1. methodology- what is the rationale for  patients with diabetes to be excluded? 

2. discussion- the fact that you did not have control group without IBD you cannot make a statement that NAFLD is more prevalent in patients with IBD.

3. changes in gut microbiota and shared pathogenesis with other gastrointestinal/endocrine disorders deserves more attention in discussion

https://pubmed.ncbi.nlm.nih.gov/33951119/

https://pubmed.ncbi.nlm.nih.gov/15115922/

Author Response

Point-by-Point-Response to Reviewers
We would like to thank both reviewers for their constructive comments and suggestions.

Reviewer 1

1. Methodology- what is the rationale for patients with diabetes to be excluded?

Response: The aim of our study was to identify factors associated with hepatic steatosis which are related specifically to chronic inflammatory bowel diseases.
Therefore, we excluded several factors that are well known to contribute to the pathogenesis of hepatic steatosis such as increased alcohol consumption and diabetes (Leite NC, Villela-Nogueira CA, Cardoso CR, Salles GF. Non-alcoholic fatty
liver disease and diabetes: from physiopathological interplay to diagnosis and treatment. World J Gastroenterol. 2014 Jul 14;20(26):8377-92.)

2. Discussion- The fact that you did not have control group without IBD you cannot make a statement that NAFLD is more prevalent in patients with IBD.

Response: We agree with the reviewer that the uncontrolled design of our study precludes firm conclusions regarding the relative prevalence of NAFLD in IBD patients compared to other diseases and the general population. Nevertheless, we
wanted to put our findings in perspective. To address the reviewer ́s comment, we have deleted the following statement on page 6 of the previous version of the manuscript: “Consistent with our findings (30.3% by B mode ultrasound and 31.5% by CAP measurement),...”.

3. Changes in gut microbiota and shared pathogenesis with other gastrointestinal/endocrine disorders deserves more attention in discussion.

https://pubmed.ncbi.nlm.nih.gov/33951119/
;
https://pubmed.ncbi.nlm.nih.gov/15115922/

Response: We would like to thank the reviewer for this suggestion. We now cite the 2
papers that were suggested by the reviewer and have added the following 2 sentences in the discussion (page 8): Similar to IBD, there is evidence for substantial changes in microbiota in diverticular diseases of the colon (Pepercorn et al.). Colonic diverticular disease is also associated with NAFLD (Melovanovic et al.) suggesting that changes in gut microbiota may play a role in the pathogenesis of NAFLD in these conditions.”

Reviewer 2 Report

Congratulations to the authors for a very interesting, original and useful research idea for IBD patients. Before going further with the manuscript, I have some comments for the authors.

- Concerning the medication, did the authors took into account only the present moment when the patient was treated with that medication, or do they have some data regarding the history of the use of that drug. The duration of the treatment with a certain drug could maybe constitute a better predictive variable.
- The authors did not mention whether any of the patients had a history of Sulfasalazine use, as some studies suggest a degree of hepatotoxicity (https://www.ncbi.nlm.nih.gov/pmc/articles/PMC2329632/). If yes, a comparison between groups would be needed.
- It is not mentioned in the Methods section what test was used to determine the parameters associated with the development of hepatic fibrosis and whether adjustments for confounders have been made
- In table 1, the rows are shown at different offsets for each column and the table is hard to read.
- Regarding this phrase: "In subsequent multivariate regression analysis disease duration was the only variable that was independently associated with hepatic steatosis." A table or a figure with the exact results and p values of the multivariate regression would be more scientific and would allow us to evaluate the degree of association, coefficients. The same regarding the phrase: "Parameters associated with the development of hepatic fibrosis were disease duration, type of inflammatory bowel disease, and several liver function tests.", a table with the exact results, p values would be welcome.

Round 2

Reviewer 2 Report

Dear Authors,

Your work and efforts during revision are visible. After these minor observations, I consider the paper can be accepted for publication.

1. Maybe this aspect should also be added in the limitations section: "Unfortunately, we have no detailed information about the duration of exposure to the respective drug.

5. Maybe it would be useful to add the new tables as supplementary files.
